# Numerical and Experimental Study on the Direct Chill Casting of Large-Scale AA2219 Billets via Annular Coupled Electromagnetic Field

**DOI:** 10.3390/ma15051802

**Published:** 2022-02-28

**Authors:** Haodong Zhao, Zhifeng Zhang, Yuelong Bai, Bao Li, Mingwei Gao

**Affiliations:** 1National Engineering & Technology Research Center for Non-Ferrous Metal Composites, GRINM Group, Beijing 100088, China; zhahadngooo@163.com (H.Z.); libao@grinm.com (B.L.); gaomingwei1990@163.com (M.G.); 2GRINM Metal Composites Technology Co., Ltd., Beijing 100088, China; 3General Research Institute for Nonferrous Metals, Beijing 100088, China

**Keywords:** internal coupled electromagnetic stirring, Lorentz force, fluid flow, grain refinement, macrosegregation

## Abstract

The internal coupled electromagnetic melt treatment (ICEMT) method is firstly proposed to produce high-quality and large-sized aluminum alloy billets. A three-dimensional model was established to describe the ICEMT process of direct chill casting (DC casting). The effect of ICEMT on the fluid flow patterns and temperature field in the DC casting of ϕ880 mm AA2219 billets is numerically analyzed. Moreover, the mechanisms of the ICEMT process on grain refinement and macrosegregation were discussed. The calculated results indicate that the electromagnetic field appears to be coupled circinate at the cross section of the melt, the fluid flow becomes unstable accompanied by the bias flow, and the temperature profiles are significantly more uniform. An experimental verification was conducted and the results prove that compared with traditional direct chill casting, the microstructures of the AA2219 large-scale billet under the ICEMT process are uniform and fine.

## 1. Introduction

AA2219 has many advantages including high specific strength, high fracture toughness, high corrosion resistance, and excellent weldability, which has been broadly applied in many fields such as aerospace, defense, and military [1,2]. With the development of heavy-lift rockets, the large-sized AA2219 billets are sorely needed to manufacture a critical component called the ultra-large-scale integral ring. DC casting is an efficient way to produce large-sized Al-alloy billets but there remain a few problems to be considered. In DC casting, the molten aluminum is continuously poured into the mold and then a solidified shell is formed when the melt comes into contact with the cold graphite ring, known as the primary cooling zone. Once it becomes strong enough to hold on to the melt, the dummy block is moved down into a pit at a speed that is gradually increased to a constant speed, known as casting speed. Simultaneously, the shell is pulled out from the mold and chilled by water, and then the solidification remains in a stable condition until the casting completes. The unique solidification mode ensures the high stability of small-scale billets. However, there is a huge difference in cooling intensity between the surface and the interior of large-scale billets, leading to the observable heterogeneity of grains and other quality problems. The fine structure is formed near the billet surface but the cooling rate at the interior of the billet is much less than that at the surface. Hence, a coarser structure is formed inside the liquid sump. Furthermore, macrosegregation gets worse with the increase in billet diameter because of the severe relative movement of solute-rich liquid and solute-lean solid in the deeper sump. AA2219 itself has a few shortcomings such as high copper content and wide temperature range of solidification, from approximately 913 K to 816 K, resulting in worse problems such as coarser and inhomogeneous structures, more serious macrosegregation, and coarser second phases in the large-sized billets. Those defects that cannot be improved in subsequent processing such as forging and rolling, are very harmful to mechanical property in service [3]. Moreover, those unrecoverable defects usually become much worse with the increase in billet diameter in the DC casting process [4].

Recently, many improvements have been proposed to obtain high-quality billets. Most of them are carried out on the external fields including ultrasonic vibrating, electromagnetic stirring, or some various combinations of them [5,6,7,8]. Zhong [9] has studied the effect of ultrasonic treatment on the metallurgical structures and an AA2219 ingot with a diameter of 1250 mm was manufactured. Though it was reported that it refined the grains, more micro defects and positive segregation in the center of the billet were observed [10]. Casting with the electromagnetic field has attracted a lot of attention and the presence of the electromagnetic field has significantly improved the quality of Al-alloy billets in industrial production [11,12,13]. Forced convection caused by the electromagnetic field in a contactless way can improve the chemical composition and temperature uniformity, refine the metallurgical structures and alleviate macrosegregation in the billets. Vives [14] proposed the CREM process to refine the structure and improve the surface quality of the billets. Zhang [15] produced the Al-alloy billets of various diameters ranging from 100 to 500 mm by the low-frequency electromagnetic casting (LFEC) process. As a result, the surface quality is improved significantly and the fine and uniform equiaxed microstructure is formed. Therefore, the electromagnetic field becomes the most popular method to manufacture small-scale billets in industrial production. However, the whole solidification cannot be traced experimentally, and only the results of cast structure can be detected to prove it indirectly; the underlying mechanism has not been entirely understood yet.

The development of numerical simulation technologies has made it possible to figure out the problem by coupling multi-physical fields. M.A. Waheed [16] analyzed the effects of billet diameters on flow pattern under the thermal convention in DC casting by using the low Reynolds numbers velocity variance-elliptic relaxation (Re ν^2^-f) turbulence model and the results showed that the increase in billet diameter deepened the flow penetration, weakened the recirculating vortex, and deepened the sump depth, which means that the larger the Al-alloy billet, the more the defects in the billet. G.C. Nzebuka [17] used the low-Reynolds number turbulence model to study the thermal evolution in DC casting of an Al-4Cu alloy and concluded that the cooling conditions and casting speed similarly affected the sump depth and the melt penetration depth. D.G. Eskin [4] also concluded that for a constant casting speed, a larger billet is more prone to negative centerline segregation because of the slower cooling, deeper sump, and a thicker transition region. The effect of electromagnetic field on liquid flow transport is considered during solidification in the large-scale billet. Zhang [15] stimulated the temperature, flow velocity distribution profiles, and the sump depth by coupling the electromagnetic field in the simulation of LFEC. Vanja Hatić [18] and Božidar Šarler [19] developed a comprehensive, multiphysics, meshless, unique numerical model to simulate direct chill casting under a LEFC, and the effect of the low-frequency electromagnetic force on the temperature, liquid fraction, and fluid flow are solved with the meshless diffuse approximate method under different current densities and frequencies. Wang [20] stimulated the solidification characteristic under different phase-pulsed magnetic fields using a transient two-dimensional axisymmetric model and compared the results of the components of the Lorentz force. However, when the billet gets bigger, the skin effect gets worse in the DC casting. Ren [21] numerically analyzed the electromagnetic field in a round bloom continuous casting mold with electromagnetic stirring and showed that the uniform temperature profiles shallowed the sump evidently, but the electromagnetic force was circinate and mainly distributed along the outer billet surface because of the skin effect. It became a significant problem for the large-sized billets to have a uniform temperature distribution. Pointing at the problem, Tang [22] made full use of the skin effect and proposed the annulus electromagnetic stirrings (AEMS) method. They investigated the effects of the parameters of AEMS on fluid flow, temperature flow, and solidification of A357 aluminum alloy. Luo [23,24] proposed a method called uniform direct chill casting (UDC), a combination of the AEMS and an in-mold cooler that was set in the liquid sump. Qiu [25,26] developed an internal cooling controller with the coils placed in it and served two purposes: he uniform and isotropic melt are formed by the internal electromagnetic stirring and the cooling medium took away the melting heat. They reached a remarkable achievement in the production of high-quality billets with a diameter of 630 mm. However, when it comes to larger billets, the quality was not easy to be improved significantly by a single electromagnetic field.

To solve the problem, a new method called internal coupled electromagnetic melt treatment (ICEMT) is first proposed to produce large-scaled billets. The schematic of DC casting under the ICEMT process is shown in Figure 1. An internal annular electromagnetic generator of 140 mm in height and 340 mm in diameter is settled inside the melt during DC casting. Two sets of coil assembly which consist of three pairs of coils are housed in it. One is placed at the inner ring and the other is at the outer ring. It divides the melt into two parts in the cross section, and the bulk melt can be stirred with different parameters, respectively, so different flow patterns can be adjusted to chase the high quality. In addition, there is a cooling area at the bottom of the internal annular electromagnetic generator to adjust the melt temperature. The objective of this work is to examine the applicability of this new method, and a billet with 880 mm in diameter was verified. A three-dimensional model was established to describe the ICEMT process by the Ansys software and the effects of the ICEMT on the melt flow, temperature field, microstructure, and macrosegregation are studied in detail, and its possible affective mechanism is discussed. The research is expected to guide the casting operation of a large-sized billet of Al alloy.

## 2. Materials and Methods

### 2.1. Model Formulation

To simplify the casting process and ensure the reliability of simulation results, the model formulation is based on the following assumptions:The melt is incompressible and isotropic;The turbulence effects are approximated using the realizable k-ε turbulence model;The densities are constant in their respective phases;The effect of fluid flow on the components of the electromagnetic field is neglected;The Joule heating is not considered in the computation due to its low frequency;The displacement current is not considered.

#### 2.1.1. Governing Equation of Electromagnetic Field

To couple the alternating electromagnetic field with the fluid flow and temperature profile, the electromagnetic field of the ICEMT process can be simplified as the magnetic flux density and electric intensity which is obtained firstly by solving the Maxwell equation:

Ampere’s law:(1)$$\nabla \times \vec{E}=-\frac{\partial \vec{B}}{\partial t}$$

Faraday’s law:(2)$$\nabla \times \vec{H}=\vec{J}$$

Gauss’s law for magnetism:(3)$$\nabla \cdot \vec{B}=0$$
where $\vec{E}$ is the electric strength, V/m; $\vec{B}$ is the magnetic flux density, T; t is the time, s; $\vec{H}$ is the magnetic intensity, and A/m; $\vec{J}$ is the electric current density vector, A/m^2^.

The relationships between the field quantities of the Maxwell equation are as follows:(4)$$\vec{B}=\mu \vec{H}$$
(5)$$\vec{J}=\sigma \vec{E}$$
where μ is the relative permeability and σ is the electric conductivity. Hence, the Lorentz force, f_em_, generated by the interaction of the magnetic field and the induced current in aluminum alloy melt is calculated by the following formulas:(6)$${\vec{f}}_{\mathrm{em}}=\vec{J}\times \vec{B}$$

#### 2.1.2. Governing Equation

Continuity equation:(7)$$\frac{\partial \rho }{\partial t}+\nabla \cdot \left(\rho \vec{u}\right)=0$$

Momentum equation:(8)$$\rho \frac{\partial \vec{u}}{\partial t}+\rho \vec{u}\cdot \nabla \vec{u}=-\nabla p+\nabla \cdot \left(\mu \nabla \vec{u}\right)+{\vec{S}}_{m}$$
(9)$${\vec{S}}_{m}=\rho g+\rho \beta \left(T-T_{\mathrm{ref}}\right)+\frac{{\left(1-f_{l}\right)}^{2}}{{\left(f_{l}^{3}+\chi \right)}^{3}}A_{\mathrm{mush}}\left(\vec{v}-{\vec{v}}_{\mathrm{cast}}\right)+{\vec{f}}_{\mathrm{em}}$$

Energy equation:(10)$$\frac{\partial \left(\rho H\right)}{\partial t}+\nabla \cdot \left(\rho \vec{u}H\right)=\nabla \cdot \left(\kappa \nabla T\right)$$

Turbulent kinetic energy equation:(11)$$\frac{\partial }{\partial t}\left(\rho K\right)+\nabla \cdot \left(\rho \vec{u}\varepsilon \right)=\nabla \cdot \left[\left(\mu_{0}+\frac{\mu_{0}}{\sigma_{K}}\right)\nabla \cdot K\right]+G-\rho \varepsilon$$

Turbulent dissipation equation:(12)$$\frac{\partial }{\partial t}\left(\rho \varepsilon \right)+\nabla \cdot \left(\rho \vec{u}\varepsilon \right)=\nabla \cdot \left[\left(\mu_{0}+\frac{\mu_{0}}{\sigma_{\varepsilon }}\right)\nabla \cdot \varepsilon \right]+C_{1}G\frac{\varepsilon }{K}-\rho \varepsilon$$
where $\vec{u}$ is a velocity vector, ρ is the density, p is the pressure, the effective viscosity $\mu =\mu_{d}+\mu_{t}$, μ_d_ is the dynamic laminar viscosity, μ_t_ is the turbulent viscosity; S_m_ is the additional source term, ρg is gravity, and ρβ(T − T_ref_) is the thermal buoyancy estimated according to the Boussinesq approximation, where β is the coefficient of cubical expansion and T_ref_ is the reference temperature. The mushy zone is regarded as a porous media so that Darcy’s law can be applied. A_mush_, originally introduced by Voller [27], is the constant to calculate the drag forces against the fluid flow in the mushy region, which is taken as 1.0 × 10^5^ kg/(m^3^·s), and χ, whose value is set to 0.001, can avoid division by zero in the calculation. v_cast_ is the casting speed, H is the sensible enthalpy (J), and κ is the effective thermal conductivity that depends on the temperature (W·m^−1^·K^−1^). Irrespective of the partition coefficient of solute, the liquid fraction, f_l_, that was treated as a function of temperature by the lever law, is expressed as follows:(13)$$f_{l}=\{\begin{array}{c} \;0,\;\;\;\;\;\;\;\mathrm{when}\;\;\;T\leq T_{s}\;\\ \;\frac{T-T_{s}}{T_{l}-T_{s}},\;\;\;\;\;\;\;\mathrm{when}\;\;T_{l}>T>T_{s}\;\\ 1,\;\;\;\;\;\;\;\mathrm{when}\;\;T\geq T_{l} \end{array}\;$$
where T_s_ and T_l_ are the solidus and liquidus temperature, respectively.

### 2.2. The Numerical Model

A three-dimensional model based on the actual equipment was established and the equations were solved by the software ANSYS Fluent 19.0 (ANSYS, Inc., Canonsburg, PA, USA) using 8 threads on a Dell Precision workstation (Dell, Inc, Round Rock, TX, USA). It consists of the melt, magnet yoke and coils, internal annular electromagnetic generator (GRINM Metal Composites Technology Co., Ltd., Beijing, China), an air zone, a mold of diameter 440 mm, and a hot top of diameter 430 mm. To ensure a mesh-independent solution, three types of meshes that had 110,136, 159,258, and 236,762 cells, respectively, were used to calculate it. The temperature results showed that the maximum relative error between the last two meshes was 0.0931% and running the calculation with the mesh of 236,762 cells can verify the mesh independence of the results. The computational boundary domains, meshes partitioning of geometric model, and the relative position of the coils and the melt are shown in Figure 2, respectively. The vertical dimensions of the boundary domains are given in Table 1. To clarify the effect of the ICEMT on fluid flow and solidification in the billet, both simulations, conventional DC casting and DC casting under ICEMT, were considered comparatively. It takes about 8 h for the conventional DC casting and more than 14 h for the DC casting under ICEMT process to reach the steady state. User-defined functions (UDF) written for this work were used in boundary conditions, material properties, and magnetic fields.

### 2.3. Material Properties

AA2219 was selected to be the material of the billet in this simulation, and the requisite properties calculated by JMatpro software (Version 7.0, Sente Software Ltd., Guildford, UK) are shown in Table 2 and Figure 3. The chemical composition used in this work is 6.35 wt.% Cu, 0.32 wt.% Mn, 0.15 wt.% Zr, 0.1 wt.% V, and 0.04 wt.% Ti. In the magnetic field calculation, the air zone was assumed as an insulation medium and no ferromagnetic materials exist in the model except the magnet yoke. Both induction coils with 100 coil turns are conducted with an AC of 10 Hz frequency and 20 A intensity. The magnetic properties used in the simulation are shown in Table 3.

### 2.4. Boundary Conditions

To ensure the accuracy of flow and temperature field, a correct set of boundary conditions is crucial in the simulation. Considering the complicated heat transfer, the whole boundary condition was divided into several parts including the inlet, hot top, the primary cooling, air gap, the secondary cooling, the inner cooling, and the outlet zone. All boundaries were treated as static walls, and the thermal boundary conditions are described as Cauchy-type boundaries:(14)$$k_{\mathrm{thermal}}\frac{\partial T}{\partial n}=h\left(T-T_{\mathrm{en}}\right)$$
where h is the heat transfer coefficient, T_en_ is ambient temperature, and is set to 300 K. 

In the inlet zone, the casting temperature is 973 K, and the velocities of the 2219 aluminum alloy for the flow quantities are presented as follows:(15)$$v_{\mathrm{inlet}}=v_{\mathrm{cast}}\frac{\rho_{s}S_{\mathrm{outlet}}}{\rho_{l}S_{\mathrm{inlet}}}$$
where v_inlet_ is the velocity of the inlet, v_cast_ is the casting velocity, S_outlet_ and S_inlet_ is the area of outlet and the inlet, respectively. The hot top that is 220 mm in length is assumed as an absolute insulating material, so it is treated as the thermally insulated area on the billet surface.

In the primary cooling zone, the melt which is initially in contact with the mold 60 mm in length is gradually pulled away from the surface of the mold as it solidified to a shell because the volumetric shrinkage of the shell decreases the contract pressure between the solid shell and the mold, leaving an air gap zone eventually, which drastically reduces heat transfer. Hence, a convective boundary condition varied with the mass fraction is applied and the convective heat transfer coefficient is assumed to be written as Equation (16) presented by Sengupta [28]:(16)$$h_{p}=h_{\mathrm{con}}\times \left(1-f_{s}\right)+h_{\mathrm{air}}\times f_{s}$$
where h_p_ is the convective heat transfer coefficient of the primary cooling, h_con_ and h_air_ is the convective heat transfer coefficient of the melt contact to the mold and the air gap, and both are given as 2000 W/(m^−2^·K) and 50 W/(m^−2^·K), respectively. Moreover, the temperatures of mold and air are set to 400 K and 300 K, respectively.

The secondary cooling was divided into water jet zone and downstream zone. The water jet region, set as 100 mm in length, is influenced by two main causes, heat convection and nucleate boiling heat transfer. The transition and film boiling, are nearly ignored because the surface temperature of the billet is below the Leidenforst Point, which is above 573 K for aluminum alloy. Therefore, the heat transfer coefficient in the secondary cooling zone can be presented as follows [17]:(17)$$h=\{\begin{array}{c} \frac{27,300\left(T-273.15\right)-123,088.915}{T-T_{\alpha }},\;\;\;\;\;\;\;\;\;\;\;\;\;\;\;\;\;\;\;\;T<393K\\ \frac{94,252\left(T-273.15\right)-9,240,434.453}{T-T_{\alpha }},\;\;\;393K\leq \;T<423K\\ \frac{12,259\left(T-273.15\right)-3,058,560.867}{T-T_{\alpha }},\;\;\;\;\;\;\;\;\;\;\;\;\;\;\;\;\;\;\;\;\;T\geq 423K \end{array}$$
where h is the convection heat transfer coefficient, T is the temperature, Tα is the free stream temperature. In addition, at the downstreaming zone I and II, set as 200 mm and 300 mm in length, respectively, the heat transfer coefficient expresses as below by Suyitno [29]:(18)$$h=\frac{\left(-1.67\times 10^{5}+{\mathrm{cT}}_{\mathrm{bar}}\right)\;{\left(\frac{Q_{w}}{60,000}\right)}^{1/3}\Delta T+100{\left(\Delta T_{x}\right)}^{3}}{T-T_{\alpha }}$$
where, $c=-21.2035\;Q_{w}{}^{2}-1.15080Q_{w}+62,794$, $\Delta T=T-T_{\mathrm{water}}$, $\Delta T_{x}=T-T_{\mathrm{sat}}$, $T_{\mathrm{bar}}=\left({T+T}_{\mathrm{water}}\right)/2$, T_sat_ = 363 K, T_water_ = 293 K, and water flow rate, Q_w_ = 433 L/min.

The intercooler heat transfer was determined by the heat exchange of the cooling medium, which depends on the cooling medium flow rate. The heat transfer coefficient, h_in_, can be calculated by the following equation:(19)$$h_{\mathrm{in}}=\frac{\rho_{m}v_{m}S_{\mathrm{pipe}}C_{m}\left(T_{\mathrm{out}}-T_{\mathrm{in}}\right)}{S_{\mathrm{in}}\left(T_{w}-T_{m}\right)}$$
where ρ_m_, v_m_, and C_m_ is the density, velocity, and specific heat of the cooling medium, respectively; S_pipe_ is the area of the cooling medium pipe, S_in_ is the active surface area of the intercooler; T_out_ and T_in_ is the outlet and inlet temperature of the cooling medium, respectively; and T_w_ and T_m_ is the temperature of the generator bottom surface and the melt, respectively.

### 2.5. Experimental Method

All materials such as industry pure aluminum, pure copper, the master alloy Al-4Zr, Al-10Mn, and Al-10Ti were melted at 1123 K in a furnace. After being degassed and filtered in a holding furnace, the melt was transferred into the preheated hot top while the casting temperature was controlled at 983 K. The casting speed was 26 mm/min and the casting process was processed at a stable stage for a certain length (1050 mm). Then, the preheated generator (GRINM Metal Composites Technology Co., Ltd., Beijing, China) was set in the melt and prepared to work. Hence, both processes were processed in one billet. The liquid sump depth was measured by inserting a special probe vertically until it touched the solid–liquid interface. After homogenizing (788 K × 12 hr + 798 K × 26 hr), the billet was sliced and the samples were selected along the radial direction of this slice. In micrograph analysis, the samples are prepared carefully from the cross section of billet by grounding, polishing, and anodizing in 2.5% HBF_4_ aqueous solution and the microstructure observations were analyzed and compared through the polarized light optical microscope, a Carl Zeiss Axiovert 2000 MAT (Carl Zeiss AG, Oberkochen, Germany). Solute concentrations at certain locations on the cross section of the billet are determined by the direct reading spectrometer, Foundry-Master Pro (Oxford Instrument, Oxford, UK). Measurements were performed on two sides of one specimen, with the results being averaged. The absolute error for copper concentration was ±0.1 wt.%. The specimens were taken from an identical location with a distance of 45 mm from the periphery of the billet.

## 3. Results

### 3.1. Electromagnetic Field under ICEMT Process

Figure 4 gives the magnetic flux density induced by the ICEMT generator as the AC frequency changes from 5 Hz to 20 Hz at a height of 100 mm below the inlet of melt at a given time. It differs from normal distributions of magnetic induction intensity induced by electromagnetic stirring (EMS) outside the melt. It essentially serves a dual purpose as two electromagnetic stirrers and as a controller to regulate melt flow by changing these electromagnetic parameters. The magnetic induction intensities in the melt are divided into two parts. It should be pointed out that the magnetic flux densities are a similar distribution at all the AC frequencies but the current penetration depths vary with the change of AC frequencies. At 10 Hz and 20 Hz, magnetic flux density decreases rapidly along the radial direction when the distance from the edge of the generator increases bilaterally because of the skin effect. In the meantime, there is a larger maximum magnetic flux density near the generator surface at 5 Hz than that of the other AC frequencies. Figure 5 shows the radial distributions of the simulating magnetic flux density at different AC frequencies at different positions on the cross-section of the billet. It can be figured that the distribution of magnetic flux density in the aluminum alloy melt is non-linear with the variation of AC frequencies. Moreover, the magnetic flux density outside the generator was about 80% of that value at the area inside the generator, which is coincident with the experimental data satisfactorily.

### 3.2. Melt Flow Pattern and Temperature Fields under ICEMT Process

Figure 6 shows velocity vectors and streamline patterns in the melt at steady state during conventional DC casting and ICEMT (20 Hz, 20 A for outer coil assembly and 5 A for inner coil assembly) at a casting speed of 26 mm/min. It is in a natural convection during conventional DC casting, as shown in Figure 6a. The high-temperature melt moves along the periphery and the mushy zone of the billet, rises in the center, and a huge clockwise recirculation is formed. The ICEMT significantly changes it into forced convection. There are two vortexes on the cross-section 20 mm below the inlet as shown in Figure 6d. One is located in the generator and the other is circulating outside. The tangential velocity is the highest of the three components and is increased to over 0.1 m/s. As a result, two huge and intense whirlpools and some secondary flows could distribute the melt to the surface randomly and make a great contribution to the solute distribution and temperature field. It can be seen from Figure 6c that the upward melt flows in the center part of the sump on the vertical plane during ICEMT are much stronger than thermo-solutal convection during the conventional DC casting. The melt cooled by the cooling bottom of the generator is sucked up and then mixed up with the high-temperature melt. The solute-rich liquid is transported from the center of the sump and redistributed somewhere else, which can uniform the temperature field and the solute distribution effectively. Two vortexes in opposite directions are observed in the flow field on the vertical plane during ICEMT. These vortexes could reduce the temperature gradient difference between the periphery and center of a large-sized billet. The melt flow pattern can enhance the heat loss significantly and create more uniform temperature fields leading to the liquid sump depth decrease and shallowness in the large-sized billet. The temperature under ICEMT is more uniform than that of the conventional DC casting. The difference in temperature gradient between the center and the periphery of a large-sized billet could be promoted by ICEMT. Figure 7 gives the comparison of the temperature distribution and liquid fraction on vertical planes with and without ICEMT. There was a vast temperature variation between the periphery and center of the billet during the conventional DC casting. The superheat of the melt cannot be transported out in time, resulting in a deep liquid sump as shown in Figure 7a. The temperature profiles under the ICEMT process are more uniform due to the melt flow and the intercooling function of the generator. The temperature in the liquid sump under the ICEMT process is 50 K lower than that of the conventional DC casting, as shown in Figure 7b. It is helpful to keep liquid sump shallow. The calculated sump depths in conventional DC casting and ICEMT were 462 mm and 342 mm, which are in good agreement with the experimental measurements. 

### 3.3. The Evolution of Microstructure under ICEMT Process

The ICEMT process can refine the grain structure of the billet significantly. Figure 8 shows the microstructures of the billet under both processes. For the normal billet, the grains are much coarser and the grain size is in the range from 150 μm to 1400 μm. It shows that the maximum grains are located in the billet center. The microstructures in the billet center are not uniform. There are a few small grains, while others mostly are large with thicker dendrite arms and large dendritic arms spacing. The structure with ICEMT mainly comprises fine equiaxed grains and is generally uniform across the cross-section of the large-sized billet. Grains are in the range of 100~300 μm in the average grain size, much smaller than that of a conventional billet. Figure 9 provides quantitative information on the average grain size at the different positions under both processes and shows that the billet without ICEMT produces much larger grains as compared with the one with ICEMT. The application of ICEMT vastly reduced the average grain size at the center and periphery for the billet with ICEMT. In addition, it also seems to homogenize the grain structure at the same position, as indicated in the big reduction in the error bars on the average grain size measurements.

Experimental concentration profiles for copper on the cross section of the billet with and without the ICEMT process are given in Figure 10. The relative copper concentration, ΔC, is defined as the equation as follows [30]:(20)$$\Delta C=\frac{C_{i}-C_{0}}{C_{0}}\times 100\%$$
where C_i_ is the running concentration on the different positions and C_0_ is the nominal composition. It can be seen that the application of ICEMT does dramatically change the severe centerline macrosegregation in the billet center to slight positive macrosegregation.

## 4. Discussion

During the conventional DC casting, the melt flow in the hot top could be described as a relatively quiescent and thermally stratified fluid. When it is close to the mold wall, where the temperature decreased rapidly, the hot melt would be cooled down and flow along the solidification front. These flows were stopped at the bottom of the sump and the flows mixed with some small grains turned upwards with the help of thermal buoyancy. Most grains would be remelted and the high temperature and poor cooling rates are not allowed to nucleate in the sump, so the grains coarsen to a large columnar structure in the central part of the billet. During the ICEMT process, the grain size is more homogeneously distributed on the cross section of the billet than that of conventional DC casting. Columnar structures are vanished, which is a result of unique intensive convection and lower thermal gradients achieved by intercooling. A uniform temperature profile in the liquid sump was obtained by the ICEMT process. As result, the decrease in melt temperature shallows the sump and broadens the transition region by shifting both the liquidus and the solidus isotherms, as shown in Figure 8b. In addition, liquidus isotherms were affected to a greater extent, resulting in a shallower and more isothermal sump. More nucleation occurs at the same time in the entire volume of the melt and the uniform temperature field created a suit condition to let the nucleation grow to small equiaxed crystals.

There is more intensive transport of the small grains under the ICEMT process. The intense melt flow destroys the solute boundary layer nearby the grain, and the growth of dendrites was suppressed. At the solidification front, the fragile portions dendrites were flushed, remelted, or fractured by the intensive forced flow and then detached from the solidification front and play an important role as nucleation particles. Therefore, the fine grains are formed, grow, and then act as heterogeneous nucleation sites ahead of the solidification front. Melt flow under the ICEMT process has a significant flushing effect on the generator surface, and the cold bottom of the generator would also provide heterogeneous nucleation sites for the explosive nucleation. Consequently, these grains are largely detached into the melted center to become nucleation sites, which promote heterogeneous nucleation to refine grains. All methods would increase the nucleation rate and thus refine grains in the billet.

The intensive flow would disperse the cool and solute-rich liquid in time due to the partitioning of solute elements between liquid and solid phases during solidification. The main element alloy in AA2219, Cu, is enriched in the liquid phase with a partition coefficient, K, less than 1. Solute elements in the liquid phase generally slow down the growth velocity of the newly formed grains, prevent the formation of dendrites and more grains are thereby allowed to nucleate. Both billets prepared with and without ICEMT process have obvious segregation near to the periphery and the ICEMT process does not show significant effects on it. The periphery is not crucial for the 2219 alloy billet because it will be scalped before follow-up processing. However, the positive segregation of the AA2219 billet center is reduced and even the negative segregation is presented in the opposite by the ICEMT process. It is very significant for the large-sized AA2219 billet to control the macrosegregation. In the conventional large-sized billet, the shrinkage-induced flow played an important role to aggravate the positive segregation, but thermo-solutal convection played a decisive role in the formation of positive segregation. As for the ICEMT process, the intensive melt flow forced the solute-enriched liquid to flow out from the bottom of the liquid sump. As a result, positive segregation fails to form in the billet center.

## 5. Conclusions

A three-dimensional model that coupled the electromagnetic field, melt flow, and solidification was established to investigate the solidification behavior undergoing the ICEMT process. ICEMT was carried out in the DC casting process for the production of a large-scale billet, contrastively. The effect of ICEMT on the resulting microstructure was observed. The main conclusions from this work can be summarized as follows.

The magnetic flux density is unique and uniformly distributed under the ICEMT process. Significant improvement in the control of melt flow and temperature field can be obtained by the ICEMT process. The melt flows pattern includes two vortexes in opposite directions outside the generator and an upward whirlpool inside the generator. The temperature distribution in the melt with ICEMT is more uniform than that without ICEMT, leading to the sump being shallower by about 100 mm.

Grains are much refined from 700~1400 μm (without ICEMT) to 100~300 μm (with ICEMT) and grain structure is uniformly distributed under the ICEMT process. The application of ICEMT dramatically changes the macrosegregation on the cross section of the billet. The severe positive macrosegregation in the center was promoted significantly.

## Figures and Tables

**Figure 1 materials-15-01802-f001:**
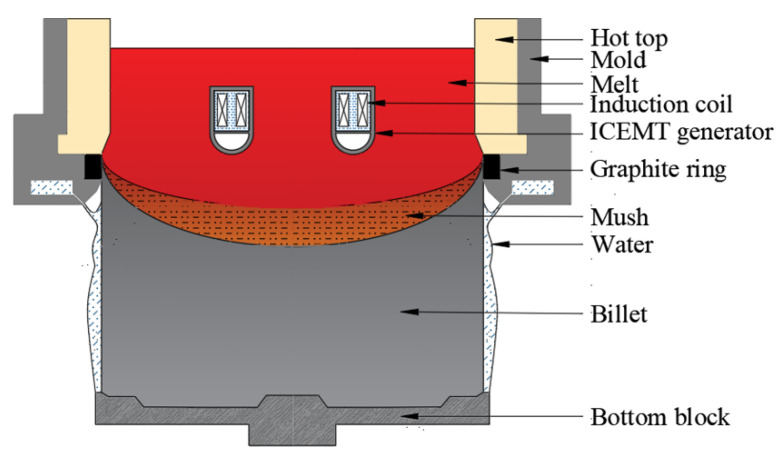
Schematic of the casting process under ICEMT.

**Figure 2 materials-15-01802-f002:**
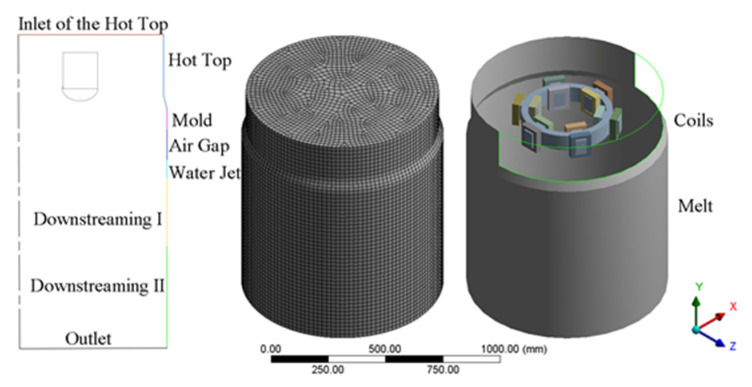
Schematics of the computational boundary domains, mesh partitioning of billet, and location of coils in the melt.

**Figure 3 materials-15-01802-f003:**
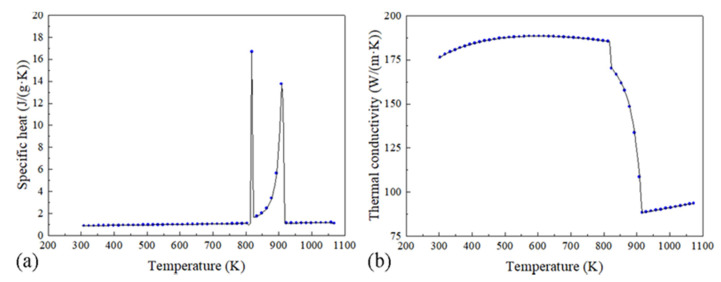
The values of specific heat (**a**) and thermal conductivity (**b**) used in this work.

**Figure 4 materials-15-01802-f004:**
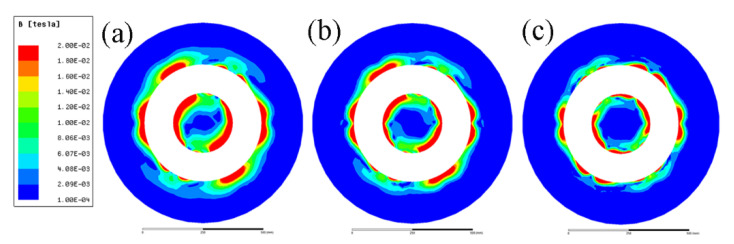
Distribution of magnetic flux density conducted with an AC frequency: (**a**) 5 Hz, (**b**) 10 Hz, and (**c**) 20 Hz.

**Figure 5 materials-15-01802-f005:**
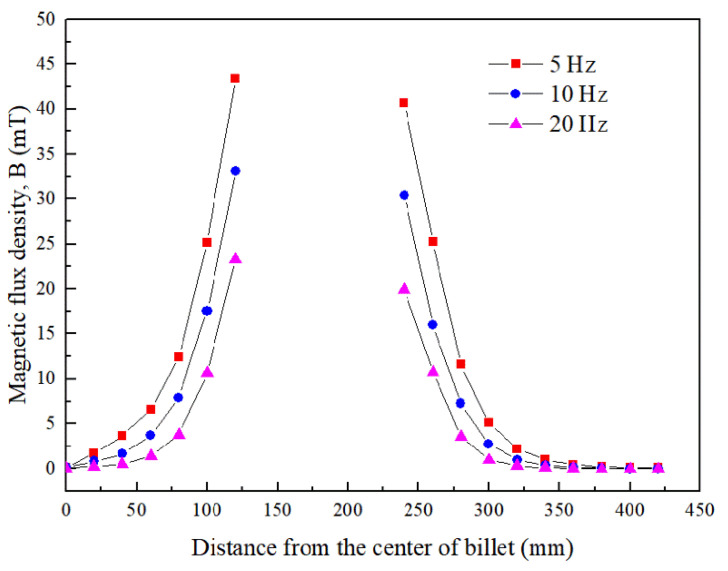
Radial distributions of magnetic flux density on the cross section of billet.

**Figure 6 materials-15-01802-f006:**
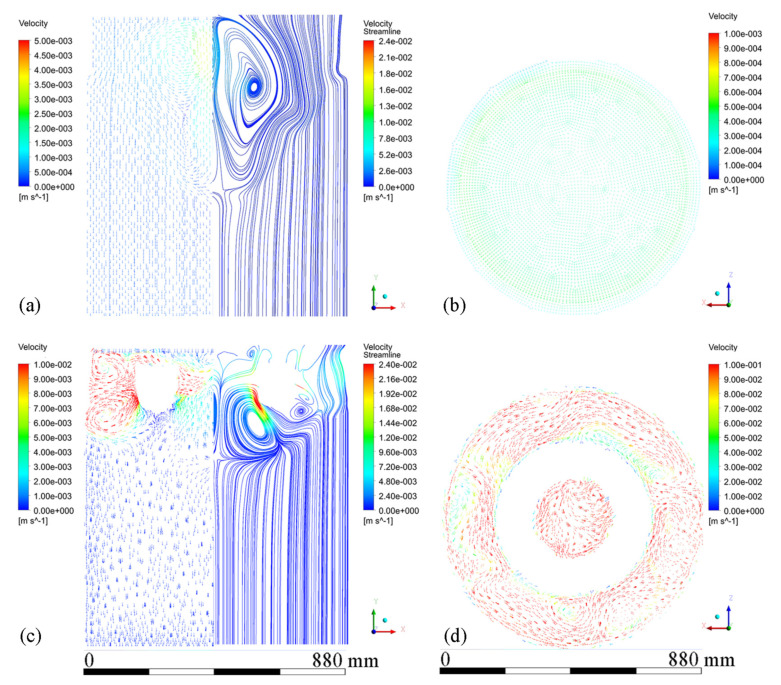
Velocity vector plots of melt flow and streamline patterns: (**a**) on the vertical plane during conventional DC casting, (**b**) velocity vectors on the horizontal plane 20 mm below the inlet level during conventional DC casting, (**c**) vertical plane ICEMT DC casting, and (**d**) velocity vectors on the horizontal plane 20 mm below the inlet level during ICEMT.

**Figure 7 materials-15-01802-f007:**
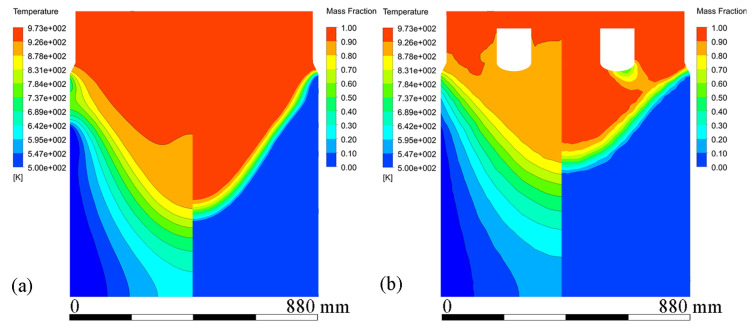
Contours of temperature profiles and mass fraction at the stable stage under both processes: (**a**) conventional DC casting, and (**b**) ICEMT.

**Figure 8 materials-15-01802-f008:**
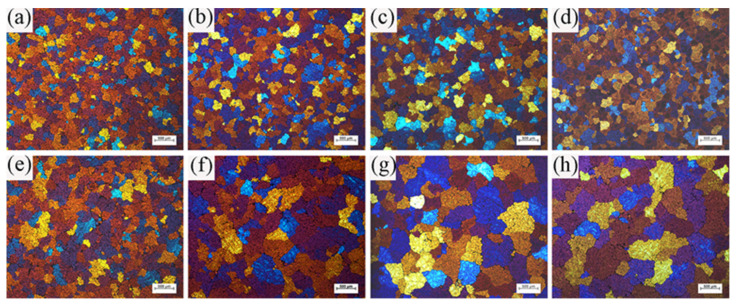
Typical grain morphologies at different positions of billets obtained under both processes: (**a**) ICEMT edge; (**b**) ICEMT 1/3 radius; (**c**) ICEMT 2/3 radius; (**d**) ICEMT center; (**e**) conventional edge; (**f**) conventional 1/3 radius; (**g**) conventional 2/3 radius; and (**h**) conventional center.

**Figure 9 materials-15-01802-f009:**
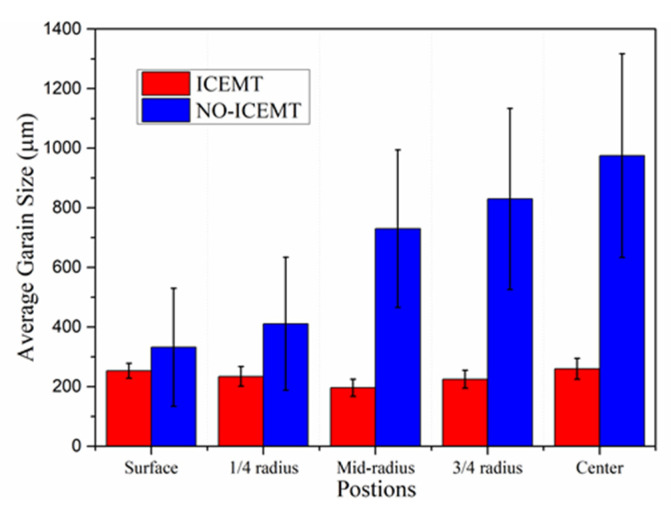
Average grain size at different positions in the billet under both processes.

**Figure 10 materials-15-01802-f010:**
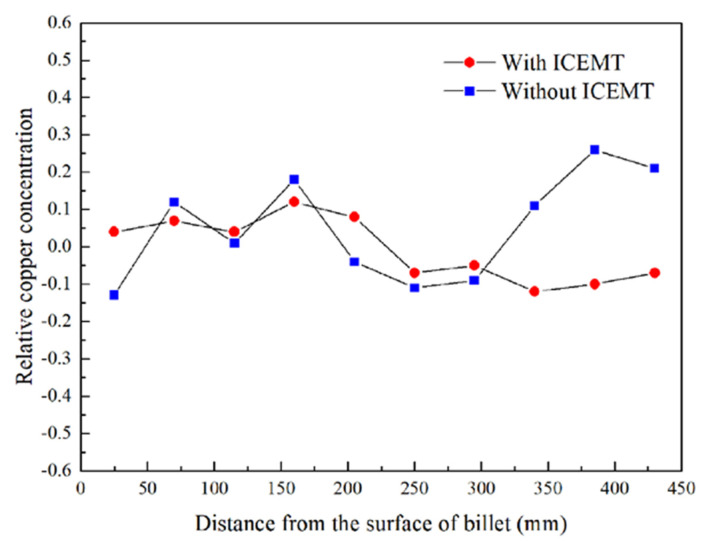
Experimental macrosegregation in AA2219 billet with and without the ICEMT.

**Table 1 materials-15-01802-t001:** The vertical dimensions of the boundary domains in the present simulation.

Geometry Dimensions	Values/mm
Hot top	220
Mold	60
Air gap	120
Water jet	100
DownstreamingI	200
DownstreamingII	300

**Table 2 materials-15-01802-t002:** The thermophysical properties of 2219 aluminum alloy.

Thermophysical Properties	Values	Unit
Liquid density	2520	kg/m^3^
Solid density	2719	kg/m^3^
Dynamic viscosity	1.42 × 10^−3^	Pa·s
Solidus temperature	813	K
Liquidus temperature	916	K
Specific heat	As shown in Figure 3a	J/(g·K)
Latent heat	3.85 × 10^6^	J/kg
Liquid thermal expansion coefficient	4.4 × 10^−5^	1/K
Thermal conductivity	As shown in Figure 3b	W/(m·K)

**Table 3 materials-15-01802-t003:** The magnetic properties used in the ICEMT process.

Physical Properties	Al-Alloy	Coils	Yoke	Air
Relative permeability	1	1	2000	1
Relative dielectric	1	1	1	1
Conductivity (S/m)	3.8 × 10^7^	5.8 × 10^7^	2 × 10^6^	0

## Data Availability

The data that support the finding of this study are available from GRINM Metal Composites Technology Co., Ltd. Restrictions apply to the availability of these data, which is used under license for this study. The data is available from the authors with the permissions of GRINM Metal Composites Technology Co., Ltd and Xinjiang Joinworld Company Limited.

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
