# Peer review of "Numerical and Experimental Study on the Direct Chill Casting of Large-Scale AA2219 Billets via Annular Coupled Electromagnetic Field"

_materials, 2022, doi:10.3390/ma15051802_

Round 1
Reviewer 1 Report
The manuscript "1572200 - Numerical and experimental study on the direct chill casting of large-scale AA2219 billets via annular coupled electromagnetic field" presents interesting results about the internal coupled electromagnetic melt treatment method using a three-dimensional model and experimental verification. The paper presents relevant information. It is well designed and easy to follow. The applied methodology is adequate and the literature cited is actualized. It deserves publication in Materials after Minor Revisions. The authors must consider the following comments:
- Rewrite the last sentence of the abstract section.
- Do not use the words in the title as your keywords. Change your keywords.
- Improve academic English through manuscript, for example in table 1.
- In table 1, put the unit in the first row (mm).
Reviewer 2 Report
The authors should correct their paper with the following amendments and answers:
The following publications might be cited, directly related to the presented research.
https://www.researchgate.net/profile/Bozidar-Sarler/publication/337358303_Multi-Physics_and_Multi-Scale_Meshless_Simulation_System_for_Direct-Chill_Casting_of_Aluminium_Alloys/links/5fc492e7a6fdcc6cc684b350/Multi-Physics-and-Multi-Scale-Meshless-Simulation-System-for-Direct-Chill-Casting-of-Aluminium-Alloys.pdf
https://www.researchgate.net/publication/320005656_Simulation_Of_Direct_Chill_Casting_Under_The_Influence_Of_A_Low-Frequency_Electromagnetic_Field
The computational time and platform should be described. Which version of ANSYS FLUENT has been used? What is the discretisation density?
How did the authors check the mesh independence of the results?
The left-hand side of Eq. 10 is wrong since the specific heat is missing! Why is the equation not written in terms of enthalpy, like in the solid phase?
The vertical dimensions where different boundary conditions are defined should be elaborated, consistent with the naming in Table.1
Where is the horisontal cross-section of the billet defined in Fig. 6.
It is not clear if the solution of the average velocity field is steady or not? This issue should be clearly elaborated.
Equation 13 is not defined for T_S and T_L..
Author Response
Many thanks for your letter. We have modified our manuscript as requested by the reviewers and addressed their comments point by point below. We hope that this will make the manuscript suitable for publication. Please see the attachment.
